# Extra Virgin Olive Oil from Destoned Fruits to Improve the Quality of the Oil and Environmental Sustainability

**DOI:** 10.3390/foods11101479

**Published:** 2022-05-19

**Authors:** Maria Teresa Frangipane, Massimo Cecchini, Riccardo Massantini, Danilo Monarca

**Affiliations:** 1Department for Innovation in Biological, Agro-Food and Forest systems (DIBAF), University of Tuscia, Via San Camillo de Lellis, 01100 Viterbo, Italy; massanti@unitus.it; 2Department of Agriculture and Forest Sciences (DAFNE), University of Tuscia, Via San Camillo de Lellis, 01100 Viterbo, Italy; cecchini@unitus.it (M.C.); monarca@unitus.it (D.M.)

**Keywords:** extra virgin olive oil, destoned, environmental sustainability, quality

## Abstract

The world production of olive oil represented 3.1 million tons in 2021 and the choice aimed at high quality extra virgin olive oils is increasingly appearing (IOC, 2022). Moreover, the production of a product of quality with environmental respect is grown in demand. Consequently, the so-called “ecological” processes mostly interest the production market of extra virgin olive oils. Despite the current processing and extraction technologies, the characteristics of olive oil can still be optimized. In this regard, interesting technology to produce olive oil remains the stone removal of the olives before the extraction of the oil. Recently, the destoners preserved a less low oil yield. In light of recent progress, the review focuses on the influence of destoning on the quality of extra virgin olive oil, using a systematic approach. Interest in this technology is increasing and many researchers report that destoned olive oils show superior characteristics confronting with those obtained by the traditional method. These data indicate that destoning is one of the most significant advantages for the improvement of the oil qualitative traits and the system’s sustainability.

## 1. Introduction

Many factors influence the quality of virgin olive oil: olives, harvesting methods, extraction technologies from the crushing of the olives to the separation of the oily phase [1,2]. All the operations required in the oil extraction process are aimed at obtaining the highest quality of oil from the fruits. In this context, the preparation phase of the olive paste is very fundamental [3]. Over time, traditional pressure extraction has been replaced with the centrifugation system; this system has some disadvantages, due to the addition of hot water to the olive paste. Therefore, a two-phase centrifugal decanter has been manufactured which can separate oil from pasta without adding water [4]. Extra virgin olive oil of high-quality presents both very special sensory characteristics and health benefits, therefore there is an increased consumers demand for this product. The goal of increasing the quality standards for virgin olive oil has stimulated the research for new technologies. Twenty years have passed since various producers have developed technological procedures that include removing the stone before the olive oil extraction process [5,6,7,8,9,10,11]. In the production plants of extra virgin olive oil from pitted olives, the pit removal machines are placed at the beginning of the olive processing line [12]. The washed olives are placed in continuous pitting machines which can be of two types [13,14]: total pitting machines (which eliminate all the stones) and partial pitting machines (which eliminate part of the stones). Total pitting machines had a limited diffusion because, despite adjustments of the decanter, the extraction yield is always slightly lower than that obtained using a paste containing a network of stone fragments. However, this problem is now reduced by controlling the water content of the pitted pulp, to ensure optimal separation of the oil [15,16]. The pitting takes place owing to the action of a rotating shaft (700–800 rpm) connected to metal bars coated in rubber, placed inside a horizontal perforated cylinder (holes from 4 to 6 mm) in turn placed inside a cylindrical casing with a continuous wall with, underneath, a tank for pulp collection (Figure 1).

The olives enter the machine by means of an auger. The pulp comes out through the holes in the internal cylinder and falls into the collection tank, while the rotating shaft pushes the stones outwards, on the side opposite the feeder. The hourly capacity of the machine is about 2000 kg/h of olives and the pulp does not undergo significant temperature increases [13]. The partial pitting machines (Figure 2) consist of two sections [16,17]: In the first, normal milling takes place through two counter-rotating toothed rollers at different angular speeds (about 70 rpm and about 140 rpm); the second section is very similar to a total pitting machine, with the difference that the holes in the internal cylinder have smaller diameters (2.5–3.5 mm). In this way, some stone fragments remain in the paste (generally 50–70%). The hourly capacity of these machines varies between 2000 and 6000 kg/h of olives [13]. With particular varieties of olives or with olives in the most advanced state of ripeness, pitting can cause some problems: in fact, stone dust can be generated which can clog the decanter, reducing the oil extraction yield [18]. The main advantage of using the destoned paste is that it ameliorates the sensory properties and prolongs the shelf-life of extra virgin olive oil. In the destoning process, the enzymes which were contained in the seeds are turned away so that they do not catalyze the oxidative reactions of chemical compounds [19,20,21].

Moreover, thermal activities, responsible for the decomposition of different constituents present in oils, had a decrease [22]. In some research [23,24], it was observed that the destoning technique noticeably affected the phenolic compounds, in fact the destoned fruit oils were characterized by higher phenols content. These findings showed that owing to the peroxidase (POD) activity observed, the seed influences the phenols oxidative reactions, above all in the extractive phase [25]. Ranalli et al. [26] investigated the oils from destoned olives compared with non-stoned olives. Stone removal allowed the production of highly nutraceutical oils, rich in biophenols. Results reported in the literature [3] showed that the oil of destoned fruits had inferior values of free acidity and the spectrophotometric indices (K232 and K270, which show UV absorption in 232 nm and 270 nm) than the oil obtained by traditional methods, therefore, they had less oxidation. Both the phenolic fraction and the volatile compounds increase in olive oils obtained from destoned olives, leading to an improvement in the nutritional and sensory characteristics of the product. It was observed that stone removal increased in the oils the volatile compounds bonded to the “green” probably because the enzymes involved in the lipoxygenase (LOX) pathway led to different products depending on whether they are found in the pulp or in the seed [11]. Titouh, Mazari and Meziane [27] reported the effect of both destoning of olives and the addition of talc as co-adjuvant, used in modern oil mills, during malaxation of the paste on the yield of extracted oil. As expected, destoning carried out a slight decrease of about 1.1 percent point of the oil yield compared to the whole fruits. It was observed that the addition of talc at 2.5% does not significantly improve the oil extraction from destoned fruits; however, the destoning of olives allowed improvement of oil quality, valuing local the traditional olive-growing by giving them bonding to the territory. In this sense, the authors noted in fact an enhancement in qualitative traits of oil which could add value to the local products distinguishing in comparison to those obtained by standard methods. In another study conducted by Guermazi, Gharsallaoui, Perri, Gabsi and Benincasa [28], the life cycle analysis was evaluated in a new olive oil technology process, consisting of a destoning and a two-phase extraction system. They obtained both the pulp to produce quality extra virgin olive oil and the stones to produce energy, thus concurring to reducing the environmental impact. Romaniello, Leone, and Tamborrino [16] designed and built an industrial prototype of a partial destoner machine. This machine did not completely remove the stone, but only a quantity of about 50% of the olive stone fragments. The authors investigated the extraction efficiency of the implant, the quality of the olive oil, and the rheological aspects. The results pointed out that the partial destoner machine compared to the total destoning allowed an increase in the extraction yield, a significant reduction in the viscosity of the paste, and the stones can be recovered. Moreover, the oils from the partial destoner machine were distinguished by the intensity of fruity flavor and aroma in comparison to samples from whole olives. A recent work, particularly interesting [17], introduced a new partial destoning machine (called Moliden), which had been placed in an experimental trial to assess the impact on the quality and sustainability of the extraction process. The partial de-stoner machine proposed by the authors allows considerable savings in the production process since it includes two sections: crushing and destoning. This aspect is especially important since compared with the standard parameter of the high-quality wood pellet it provides a higher quality stone to be used as a biomass fuel. This could have a significant impact on the environmental sustainability of the process. The destoning treatment also played an important role in the improvement of the aldehydes and esters with a positive impact on extra virgin olive oil flavor. Moreover, the authors found an increase in bioactive compound content that enhanced bitter and pungent sensory notes. Yorulmaz, Tekin and Turan [29] evaluated the influence of stone removal and malaxation in the nitrogen atmosphere on the oxidative stability of the oils. The findings highlighted that the combined effect of malaxing under nitrogen and destoning made it possible to obtain high-quality oils. Although the employment of the destoner can ameliorate the working capacity of the mill plant, as it eliminates part of the solid waste before extraction, a third-generation decanter is needed to separate the oil from the olive paste, since the removal of the stone changes the rheology of olive paste [30]. Other authors [31,32] have shown that de-stoner technology could represent a useful sustainability tool for olive oil extraction plants. In particular, stones of the olives can be used as fuel allowing significant energy savings [33]. In a recent study [34], the nutritional characteristics of destoned olive oils have been considered. The authors concluded that destoning technology could enhance both the sensory characteristics and nutritional value of the oil. The present work revisits, in light of recent progress, the state-of-the-art of the influence of stone removal on the quality of extra virgin olive oil with particular emphasis on phenols, volatile compounds, and sensory characteristics. It is hoped to give new life to destoned technology, with a significant advantage for the quality of the extra virgin olive oil and the sustainability of the system. It is now evident, that the interest in olive oil from stone removal is growing, since this technology allows a better working capacity, decreases waste generation, and improves virgin olive oil quality. This review will help olive oil producers to do the best choice for enhancing the qualitative characteristics of the product. An ulterior goal of our work is also to represent an important resource for scientists. It can offer inspiration for one’s own research to other researchers. Finally, it is noteworthy to report that in destoned olive oil production there are by-products with a lower environmental impact. In fact, due to the stones being about 25% of the total olive paste volume, with the stone removal, the solid waste processing quantity is considerably lesser [32].

## 2. Fruit Characteristics Affecting Destoning

The virgin olive oil composition depends primarily on olives characteristics, bonded on many parameters such as cultivar, ripening phase, and environmental conditions. An important role is played by the size of the olives; large olives are more suitable for the destoner machine. In fact, if the pulp is low, an excessive pulverization of the stone occurs. This, by increasing the adsorbing capacity, causes the loss of the oily fraction. It follows that the pulp must be thick and the stone small [35]. Kartas et al. [36] showed that the genetic features of a variety have a significant impact on the pulp/stone ratio. Varieties with a high fruit weight had a high pulp/stone ratio (8.25 to 6.07), while those with small fruits had a lower pulp/stone ratio (4.40). An experiment was carried out to study the influence of different amounts of irrigation water to olive trees of Coratina and Dolce cultivars [37]. The authors found that the features of olive trees were mostly influenced by irrigation; thus, the pulp/stone ratio gradually decreased with a lowering amount of water during irrigation. Morales-Sillero, Fernández and Troncoso [38] studied different doses of N-P-K fertilizer, and its effect on nutrient concentrations, yield, and oil quality. The fruit weight and pulp/stone ratio increased with fertilizer dose. However, olive oil quality was negatively affected by increasing fertilizer: polyphenol total content, bitterness, oxidative stability, and the relation of monounsaturated/polyunsaturated fatty acids decreased. Rosati, Caporali and Paoletti [39] observed that olive trees treated with N and K fertilization showed an increase in fruit weight and pulp/stone ratio. Another study evaluated how all qualitative characteristics of olive oil were influenced by foliar application with magnesium and potassium. Results showed that the pulp/seed ratio of olive fruits significantly increased after the treatments [40]. As regards the harvesting methods, many harvesting methods exist, and the choice depends on many factors [41]. Hand-picking is the slowest and most expensive method, but it allows the producer to get the best quality of the fruit. Ahmad [42] studied the performance of mechanical systems for the olive harvest and how they influenced the quality of the final product. Results revealed that mechanical harvesting increased productivity, but caused higher percentages of fruit damage, with respect to that obtained when the olives were manually harvested. However, divergences between the data reported in several studies have been found. Some researchers [43] compared different systems of harvesting olives from the tree (manual and mechanical). These studies have shown that mechanical harvesting has achieved the best results, with labor and time savings. Mechanically harvested olives were more intact than those harvested with other systems and have produced oil of high quality. The right choice of harvesting method is very important and must be taken not to damage the olives. This is one of the key points for obtaining good results from the use of destoner technology. In this regard, it is important to underline that, the damages caused to fruits due to errata harvesting have harmful effects on endogenous oxidative enzymes and negative consequences on oil quality. In all these cases, the enforcement of the destoning cannot assure the mentioned positive effects [34]. Based on the considerations made, destoning technology should be supported by suitable agronomic practices.

## 3. Importance of the Olives Endogenous Enzymes

To evaluate the impact of the de-stoner technology on the olive oil quality, the knowledge of how the olive’s endogenous enzymes act represents a piece of indispensable information [25]. Olives are made up of the exocarp or peel, the mesocarp or pulp, and the endocarp or fossa. There are numerous studies concerning enzymes, including lipase, peroxidase, glycosidase, lipoxygenase, and polyphenol oxidase [44,45,46]. The effects of the use of de-stoner are linked to the different sharing of enzymes in the various sections of the olive. The presence of peroxidase (POD) concentrated mainly (over 98% of the whole fruit) in the olive seed was reported in several studies [14,45,47,48]. Therefore, the exclusion of the seeds reduces the phenomena of enzymatic oxidation, especially POD oxidize main phenolic compounds, localized principally in the pulp. So, the destoned process excluding olive seeds with high peroxidase activity cut down the enzymatic degradation of phenols, and the resulting oils have higher phenol content with better oxidative stability than those obtained by whole drupes [49,50,51,52]. Polyphenoloxidase (PPO) is another endogenous enzyme of olive fruit, it is localized in the thylakoids and mitochondria. It is interesting that in the drupe mesocarp, PPO activity widely carries out its chemical activity [11,45]. However, among the enzymes, PPO plays an important role in the oxidation of phenolic substances during crushing [49]. PPO and POD can oxidize both the phenolic glucosides present in the drupe and the aglycone phenols that are formed during the processing technology to obtain the oil [48]. For this reason, the reduction of POD obtained by removing the stones had a great influence on the characteristics of the products. Lavelli and Bondesan [50] evaluated the effect of olive stone removal in six monovarietal extra virgin olive oils. The study showed that the effect of destoning was variety-dependent, and it was concluded that an acknowledgment of the endogenous enzyme heritage could be important in the management production of destoned extra virgin olive oil. Lipoxygenase (LOX) is among the endogenous enzymes of the olive fruit, the one that catalyzes the oxidation of fatty acids, in particular, linoleic and linolenic acids, with the production of volatile compounds in oils. Already about twenty years ago, some researchers [53] highlighted that LOX is mostly concentrated in chloroplasts, thylakoids, and microsomes. Generally, in extra virgin olive oils, the LOX pathway is the main cause of volatile compounds formation, responsible for fruity flavor, freshly cut grass, green fruit or vegetables such as artichoke and tomato [49]. Servili et al. [11] evaluated that the stone removal influences the LOX activity in the pastes and therefore modifies volatile composition in oils, increasing the concentration of the volatile substances, especially of hexanal, *trans*-2-hexenal, and C6 esters, with a consequent enhancement in “cut grass” and “floral” sensory notes [54,55]. Mazzuca, Spadafora and Innocenti [56] observed the two isoforms of oleuropein-degradative β-glucosidases present in the mesocarp of olives. The enzyme β -glycosidase, present in the olive pulp, is implicated in the conversion of secoridoids into aglycon forms, which demonstrated to be highly soluble in oil [57]. Clodoveo et al. [25] suggested that knowledge of the appropriate conditions of β-glucosidase activity, taking into account that the crushing system could inhibit the activity of β-glycosidases, with consequent decrease of phenols. It follows that endogenous enzymes of olives have a strong influence on the destoning process performance.

## 4. Effect of Destoning Technology on Phenolic Compounds

The interest in the phenolic compounds of olive oil is constantly growing, owing to their multiple functions, antioxidant properties, nutraceutical properties, the high stability that they provide to olive oil during storage and sensory characteristics [58,59]. Numerous studies (Table 1) have been conducted to clarify the relationship between destoning technology and the content of virgin olive oil phenolic compounds.

Servili et al. [59] highlighted the importance of phenolic constituents due to their role against the oxidation of compounds present in oils. In the paper are discussed the mechanical technologies that influence their amount in the olive oil. Among these, mechanical extraction from destoned pastes has improved the phenolic content of the oils. Total phenols (mg/Kg) of virgin olive oils obtained from destoned and control (whole fruit) pastes were evaluated at time 0 and after 12 months of storage at room temperature (25 °C). Oils of destoned olive pastes had a content of 355 at the time 0 and 195 mg/Kg after 12 months, vs. oils of whole fruit olive pastes, with values of 345 at the time 0 and 150 mg/Kg after 12 months. The destoning process consented in part to removing peroxidase activity during the extraction process, improving the phenolic compounds of oils and their oxidative stability. Lavelli and Bondesan [50] studied the influence of destoning on the content in secoiridoids and the antioxidant activity of oils obtained from the Leccino, Moraiolo, Frantoio, Pendolino, Taggiasca, and Colombaia cultivars. Results showed that destoning has grown the secoiridoids and the antioxidant activity of oils (up to 3.5 times). Extra virgin olive, from destoned olives Leccino *cv*, had high phenolic content (1241 vs. 429 mg/Kg for destoned and stoned samples respectively). Destoning caused a minor increase in the phenolic content of Moraiolo *cv* (respectively 1072 vs. 1115 mg/Kg for the destoned and stoned samples). So, the study also indicated that these effects depended on variety, assuming that the influence of stone removal was associated with endogenous enzymes. Mulinacci et al. [51] focused on comparing the phenolic compounds of 16 fresh commercial samples of extra virgin olive oil obtained from both stoned and traditional methods. In the oils from destoned olives, higher concentrations of phenolic compounds were found in agreement with their higher antioxidant capacity. Among investigated *cv*, Coratina showed values of 120 vs. 52.4 mg/L for 3,4-DHEA-EDA in oils obtained from stoned and whole fruits respectively. Peranzana *cv* showed values of 169.8 vs. 114.4 mg/L for 3,4-DHPEA-EDA in stoned and traditional samples respectively. Del Caro, Vacca, Poiana, Fenu and Piga [3] evaluated the impact of the destoning method on minor components and antioxidants in oils from Bosana *cv*. In the destoned oils was found higher shelf-life with respect to these obtained from traditional systems. Luaces, Romero, Gutierrez, Sanz and Perez [47] assessed whether olive seeds played a role in the phenols of olive oils. The results showed an increase in total phenols in stone removal samples produced by Spanish cultivars (Manzanilla, Hojiblanca and Picual). The authors indicated that olive seeds carry the major peroxidase activity (72.4 U g (−1) FW), responsible for the degradation of phenols. Therefore, olive seeds are fundamental in determining the phenolic profile associated with their high peroxidase activity. The increase in total phenols was noted to be superior in Picual (34%) with respect to in Manzanilla and Hojiblanca (18%). Servili et al. [11] observed in the oils from destoned olives a notable increase of the phenolic heritage, particularly the secoiridoid derivatives such as the dialdehydic forms of elenolic acid linked to (3,4-dihydroxyphenyl) ethanol and (phydroxyphenyl)ethanol (3,4-DHPEA-EDA and *p*-HPEA-EDA, respectively) and the isomer of the oleuropein aglycon (3,4-DHPEA-EA) while the lignans have not undergone any changes. The stone removal process affects especially phenolic composition in Coratina *cv*, the secoiridoid derivates such as 3,4-DHPEA-EDA shows significant modifications (365.2 mg/Kg in traditional and 507.1 mg/Kg in destoned oils). Ranalli et al. [60] evaluated the oils obtained from destoned olives (Gentile di Chieti, Caroleo, and Coratina cultivars) compared to those with traditional extraction. The destoning has made it possible to obtain highly nutraceutical oils, with a higher content of hydrophilic biophenols. Coratina *cv* showed the highest content of secoiridoids (56 vs. 44 mg/Kg in the destoned and traditional samples, respectively); Caroleo *cv* showed 39 vs. 28 mg/Kg in the destoned and traditional samples, and Gentile di Chieti *cv* had 56 vs. 44 mg/Kg in the destoned and traditional samples. Ranalli and Contento [24] evidenced the effect of the destoning technique on the concentration of bioactive compounds. They found significantly increased content in oxidized oleuropein and ligstruside derivates in two oils obtained by Leccino *cv*, destoned vs. stoned. In particular, oleuropein aglycon, dialdehydic form (9.2 vs. 4.3 mg/Kg in destoned and stoned samples respectively), and ligstroside aglycon, dialdehydic form (24.1 vs. 10.0 mg/Kg in destoned and stoned samples respectively). This increase has been attributed to the removal of the stone rich in polyphenoloxidase enzyme, which is the principal cause of phenols’ oxidative phenomena in the oils. Hence, the destoning technique ensures a higher concentration of biophenols, and also richer quantity in α- and γ-tocopherol and in α- and γ-tocotrienol, entertained as important substances having a biological effect. Discrepancies among data concentration of total tocopherols after destoning have been reported. An increase of 4–27% in the findings of some authors [3,23] and a 1–12% decrease for other ones [10,29,50] were reported. Amirante, Clodoveo, Dugo, Leone and Tamborrino [7] considered the characteristics of Coratina *cv* oils both from destoned and in whole olives. This research confirmed that the destoning process determined higher total phenols in olive oils obtained. Phenolic content was 399 mg/kg in the oils obtained from destoned olive pastes and 235 mg/Kg in those from the traditional techniques. Analogous findings were also obtained by Gambacorta et al. [10]. In their research, the total phenols in oils from Coratina *cv* were found to be higher in destoned samples (450.7 mg/Kg), versus those from whole olives (338 mg/Kg). Concerning the phenol compounds, a significant difference was highlighted mainly for hydroxytyrosol and, also in this study, stone removal produced samples rich in phenols in comparison to these obtained from the traditional process (3.28 vs. 1.65 mg/Kg in destoned and whole oils respectively), (+)-1-acetoxypinoresinol (10.55 vs. 8.43 mg/Kg in destoned and whole oils respectively), and 3,4-DHPEA-EA (10.02 vs. 7.49 mg/Kg in destoned and whole oils respectively). Restuccia et al. [21] conducted an investigation into the influence of destoning on the antioxidant properties of extra virgin olive oil from Cerasuola *cv*. The comparison of the total phenolic content of the destoned and whole samples confirmed an increase in oils from destoned pastes. The amounts were 2.65 and 1.53 μmol GA/g polar extract for fractions from destoned and non-destoned respectively. Yorulmaz, Tekin and Turan [29] analyzed the effect of stone removal with malaxation in nitrogen atmosphere on the defense to oxidation of oils from Edremit yaglik *cv*. Findings demonstrated the oils destonated and malaxed in nitrogen flush had a higher total phenols content than those obtained with the same conditions but not destoned (328 vs. 282 mg/Kg respectively). Using destoning and malaxation together in the nitrogen atmosphere also increased the oxidative stability of oils (59.50 vs. 37.70% for samples malaxed under nitrogen flush destoned and non-destoned). Moreover, the authors highlighted that while destoning alone led to a 6% decrease in tocopherol concentration, adding nitrogen washing induced a 16% increase. Ranalli et al. [26] observed phenolic constituents in destoned (vs. whole) virgin olive oil from Olivastra di Seggiano *cv*. The authors investigated from 2008 to 2010 and showed an increase in the biophenols due to the destoning process by obtaining oils of elevated quality. They considered this fact probably due to the lower thermoquinonization of the phenolic molecules and the lower activities of oxidoreductase, present in greater quantities in the stone. Total ligstroside derivatives were 98.92 vs. 83.51 mg/Kg tyrosol for destoned and whole samples respectively. Destoned oils had also higher oleocanthal levels, *p-HPEA-EDA*, compared to whole olive samples (23 vs. 21.4 mg/Kg tyrosol). Katsoyannos et al. [20] investigated the effects of stone removal with different varieties (Greek varieties Koroneiki and Megaritiki) on the phenols of oils. The phenols of oils from destoned pastes were greater with respect to these obtained by the traditional method. Moreover, the total phenol content of Koroneiki from destoned samples (303.45 vs. 226.49 mg/Kg in destoned samples and whole samples respectively) was found to be significantly greater than that in Megaritiki pitted olive oils (258.05 vs. 222.99 mg/Kg in destoned samples and whole samples respectively). It was concluded that the pitting technique maintains high content of bioactive compounds. These results were recently confirmed by Criado-Navarro et al. [61], who evaluated the effects of stone removal before processing on the bioactive constituents in virgin olive oil. In their work, they analyzed “Arbequina” and “Picual” cultivars. Destoning has been demonstrated to have different effects for cultivars and especially on secoiridoid derivatives. It was observed that these compounds in “Arbequina” oil were present in minor quantities if stone removal was conducted, while an increase was observed in “Picual” oil obtained from destoned fruits. Thus, the metabolism of secoiridoids particularly β-glucosidases and esterases resulted conditioned by the destoning of olives in a significant way. Enlightenment for this fact would be that “Arbequina” oils from destoned olives had lost enzymes which are contained in the stone. The opposite consequence was revealed for flavonoids; in fact their concentration was enhanced in “Arbequina” oil from stone removal whereas in “Picual” flavonoids decreased in virgin olive oil from destoned olives. The authors established a direct incidence on the healthful effect bonded to the phenols; stone removal contributed to decreasing the health benefits of olive oil “Arbequina” while not “Picual” *cv*. These findings were confirmed by other studies [62] that observed destoned olives showed an improvement of extra virgin olive oil quality both for phenolic and volatile composition with a significant enhancement of sensory characteristics. The authors reported the influence of ultrasound technologies on the quality parameters and sensory profile of extra virgin olive oils extracted from whole and destoned olives of the three main Italian cultivars. In the studied cultivars, Canino, Coratina, and Peranzana there was an improvement of total phenols of oils extracted from destoned olives compared to the control test, with significant increases of 21, 19.7, and 15.8%, respectively. In this regard, it was highlighted that the destoning process increased oleuropein and ligstroside derivatives. The application of ultrasound coupled with the destoning process produced a slight enhancement of phenolic concentration.

## 5. Effect of Destoning Technology on Volatile Compounds

Volatiles are very important compounds due to their impact on the flavor and sensory characteristics of oils. Many factors influence the volatile components such as cultivar, ripeness, geographic, and technological factors [54]. However, a significant proportion of the volatiles derives the enzyme activity lipoxygenase, especially during crushing and malaxation [63]. Angerosa, Basti, Vito and Lanza [5] observed that volatile components deriving from the lipoxygenase pathway were very influenced by the destoned process. In fact, volatile compounds of destoned oils from Coratina *cv* manifested a higher amount of C6 metabolites than those obtained with the traditional method. The sum of all C6 compounds, expressed as ppm, was 54.4 and 33.7 in destoned and whole samples, respectively. In particular, *trans*-2-hexenal was the main metabolite accumulated (44.8 and 30.7 ppm in destoned and whole samples). The authors concluded that the larger amount of C6 constituents in destoned oils would be related to a higher release of the membrane enzymes involved in the LOX pathway, due to effective grinding of the pulp tissues. In Coratina *cv*, Amirante, Clodoveo, Dugo, Leone and Tamborrino [7] reported an investigation on the volatile components of destoned oils in comparison with the oil from whole olives. Experimental data highlighted that the oils from destoning a greater quantity of C5 and C6 formed and were characterized by the presence of intense flavor notes in confront to these from the whole paste. Especially, *trans*-2-hexenal (185.4 vs. 110.8 mg/Kg in destoned and whole samples, respectively) and cis-3-hexen-1-ol (4.8 vs. 8.6 mg/Kg in destoned and whole samples). Servili et al. [11] investigated how destoning influenced volatile components in Frantoio and Coratina oils. The results indicated the C6 aldehydes quantity, such as trans-2-hexenal was higher in the crushed pulp, while C6 alcohols were greater in the seed. The trial demonstrated that the LOX pathway embroils diverse enzymes in various parts of the drupe. In fact, the seed showed a shorter hydroperoxide lyase and a greater alcohol dehydrogenase activity, in comparison to the mesocarp. These findings were in accordance with Luaces, Pérez and Sanz [64]. Moreover, C6 aldehydes amounts were higher also for destoned samples, whereas in the oils from whole olives the level of C6 alcohols was higher. The consequence of this feature is that the C6 unsaturated aldehydes positively enhance the cut grass notes of olive oil. Destoned oils from Gentile di Chieti, Caroleo and Coratina *cv* were compared with those obtained from whole olives [60]. The sample from stone removal showed greater amounts of volatile compounds, such as green aromas C6 aldehydes, alcohols, esters, and C5 compounds. Moreover, of great interest was the discovery of α-copaene and α-muurulene volatile components in destoned samples, in agreement with Saitta et al. [22]. Runcio, Sorgonà, Mincione, Santacaterina and Poiana [65] analyzed the influence of stone removal on volatile compounds in Carolea and Ottobratica oils. Data indicated that the destoned oils by the two morphologic different varieties had a greater content of C5 and C6, in comparison to whole oils, demonstrating that this characteristic was variety independent. In particular, the sum of the C6 compounds in Carolaea *cv* was 27.00 vs. 15.09 mg/Kg for destoned and whole samples, while in Ottobratica *cv*, it was 14.87 vs. 4.24 for destoned and whole samples, respectively. Composition of volatile fraction in destoned and whole Nocellara del Belice *cv* olives was reported by Ranalli and Contento [24]. Results suggested that destoning samples had higher concentrations of C5 and C6 volatile, responsible for the pleasant aromatic green notes in the oil. Unsaturated aldehydes were major metabolites, especially *trans*-2-hexenal was 425.1 vs. 331.2 mg/Kg in destoned and whole samples, respectively. This positive effect has been attributed by the authors to the milder functioning with small amounts of thermal energy, so as not to interrupt the hydro-peroxidelyase enzyme. Ranalli et al. [26] reported the differences of volatile composition in Olivastra di Seggiano oil obtained by destoning and traditional process. Stone removal from the fruit before processing displayed higher levels of C6 green volatiles, such as trans-2-hexenal (963.1 vs. 658.9 mg/Kg in destoned and whole samples, respectively). Moreover, in the destoned oil, the presence of new volatile molecules, α-copaene and α-muurulene, not present in the oil extracted from whole olives, was confirmed. More recently, Manganiello et al. [62] also found a growth in the total aldehydes, especially of the trans-2-hexenal for destoned oils in comparison with oils from traditional methods. The oils from Canino, Coratina, and Peranzana showed a different increment in aldehydes (33.4%, 19.4%, and 13.8%, respectively). Overall, an improvement in the quality of the pitted oils was confirmed also due to the increase in the volatile component. Several kinds of research on the influences of destoning technology on virgin olive oil volatile compounds are summed in Table 2.

## 6. Impact of the Destoning on Sensory Characteristics

Sensory traits of extra virgin olive oil depend primarily on the phenolic heritage of olive cultivars and then on many other parameters, such as technological process. As highlighted by many authors [7,10,11,24,50,51,59,60], the stone removal modifies the phenolic concentration in virgin olive oil, and consequently, also influences the sensory notes. Sensory analysis of destoned oils from Gentile di Chieti, Caroleo and Coratina cultivars was conducted, in comparison with the oils produced with the traditional process [60]. Data reported destoned oils had a delicate and harmonic flavor, characterized by marked green fruitiness, with respect to the oils obtained from the traditional system. Moreover, these positive sensory notes had scored higher by panelists (Coratina *cv* oils obtained a sensory scoring of 8.1 vs. 7.6 in destoned and traditional samples, respectively). The stoned oils had no marked bitter and astringent notes. This fact is favorable particularly for Coratina *cv* oils since it is a variety distinguished for the high intensity of bitterness and pungency. These results were in accordance with Gambacorta et al. [10]. They compared stoned and whole Coratina oils by evaluating the sensorial analysis. Samples obtained with the traditional process showed high notes in bitterness and pungency in comparison to destoned oils. Destoned oils, which had a malaxation presented a greater sensorial appreciation since they are very equable and fruity oils. Ranalli and Contento [24] confirmed that oils obtained from destoning technique were scored more by the panelists, and had a higher fruitiness, in comparison with whole olive oils. Ranalli et al. [26] reported the sensory characteristics of Olivastra di Seggiano oil from destoned olives compared to those from the traditional process. The sensory profile of the oil showed that among the sensory attributes, the artichoke taste was the most evident, and in the destoned samples, this sensorial attribute was higher compared to control (the median values of the artichoke attribute were 5.7 vs. 5.0 in destoned and whole oils respectively). In destoned oils, also the fruitiness flavor was more than the whole oils (the median values of the fruitiness were 8.6 vs. 7.0 in destoned and whole oils, respectively). Guermazi, Ghasallaoui, Perri, Gabsi and Benincasa [19] evaluated the sensory characteristic of oil produced from the whole and stoned olives of Chemlali cultivar, using the IOOC standard profile sheet method [66]. Results showed that the fruity positive attribute in conventional oil presented the lower values (2.6) while in destoned oils was almost double (5.1). For the pungent note, it was noted that the values had the same trend, they were respectively 5.7 and 2.3 in stoned and conventional oils. The stoned samples had lower values (2.1) for the bitter attribute in comparison with the conventional oils (5.7). So, the stone removal process increased the fruity and the pungent attributes, whereas it decreased oil bitterness. This fact improved the sensory quality of olive oil. In recent work [16], a partial destoning and a whole process were confronted to evaluate olive oil quality and its sensory characteristics. Data showed that the positive attributes (fruitiness, bitter and pungent) in Coratina *cv* oils obtained from partial destoned and whole olives were different. Particularly, the partial destoned oil had the fruitiness attribute higher than oils from whole olives (fruity was 3.9 vs. 2.5 respectively). Furthermore, green fruitiness and green almond were present in the partial destoned oil, while the whole oil had ripe fruitiness and ripe almond. Overall, the partial destoned samples were more harmonic in comparison with those produced using the traditional process, and for this reason, they were in better acceptance by consumers. Manganiello et al. [62] analyzed sensorial profiles of Peranzana, Coratina, and Canino cultivars highlighting that only for the Coratina cultivar, the destoning coupled with ultrasound treatment has decreased in bitterness and increased in herb sensation. In this regard, the increase in the “green” sensation contributed to enhancing the positive notes of the oil taste and the higher quantity of phenolic and volatile compounds due to the removal of the seeds, have given oil a greater sensorial score.

## 7. Future Perspectives and Conclusions

As highlighted in a previous section, the destoning of the olive paste reduces the yield of oil compared to oils from whole olives. Indeed, the destoned process generally causes a reduction of the yield to about 1.5 kg of oil per 100 kg of olives compared to traditional methods [7,14]. In the future, this negative effect could be overcome by emerging technologies such as ultrasounds [62,67]. From its application coupled with destoning, arises the opportunity to increase the oil yield without altering the nutraceutical profile [68]. In fact, the use of this emerging technology could meet the growing consumer attitude for high-quality extra virgin olive oil with health and sensory properties bonded to a higher content of phenolic and volatile compounds. An interesting future perspective is also proposed by Toscano et al. [69] who have developed a newmade lab-scale prototype to project a “low-speed” olive destoning process as an alternative to percussion and centrifugal projection for the separation of pulp stones. This method has been founded on constriction of the drupes and extrusion of their pulps by pressure. Further investigations are required to assess whether it will be possible to develop real-scale industrial destoning, evaluating all qualitative parameters of oils obtained in comparison to the current destoned method. The analysis of the literature revealed that the destoned process caused improvements in the quality of extra virgin olive oils, also owing to higher polyphenolic content. In this consideration, it is important to convey to consumers the nutritional and health added value that pitted olive oils have also by virtue of their high content of polyphenols. This fact is very important because if extra virgin olive oil has a higher phenol amount it would allow adding the health claim on the label, with a positive effect on consumers. Precisely with regard to polyphenols, the European Food Safety Authority has adopted in the Commission Regulation no. 432/2012 a series of claims regarding the advantages of constituents in foods having a biological effect, such as olive oil polyphenols. “The claim may be used on labels, only for olive oil which contains at least 5 mg of hydroxytyrosol and its derivatives (e.g., oleuropein complex and tyrosol) per 20 g of olive oil”. As pointed out, the destoning technology improves also the olive oil flavor, both for the greater concentration of volatiles answerable for the “green” sensory notes and the reduction of the bitter taste. Thus, destoned oils receive higher scores from those who particularly prize the harmonic aroma of the oils. In addition, an ulterior advantage of using destoning, which could be very important, is considering that stones can make use as biomass. At present, the purpose of environmental sustainability is increasingly towards the employment of renewable energy, and this consideration makes it very actual to re-evaluate the use of destoning technology. In our opinion, the valorization of dried stoned pomace as supplements in animal feeding seems a further point in favor of the destoned process.

## Figures and Tables

**Figure 1 foods-11-01479-f001:**
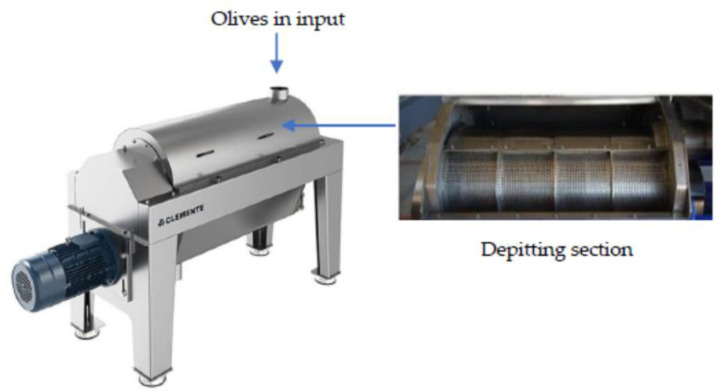
Destoning machine (photos courtesy of Clemente Industry, Olive oil Srl, Italy).

**Figure 2 foods-11-01479-f002:**
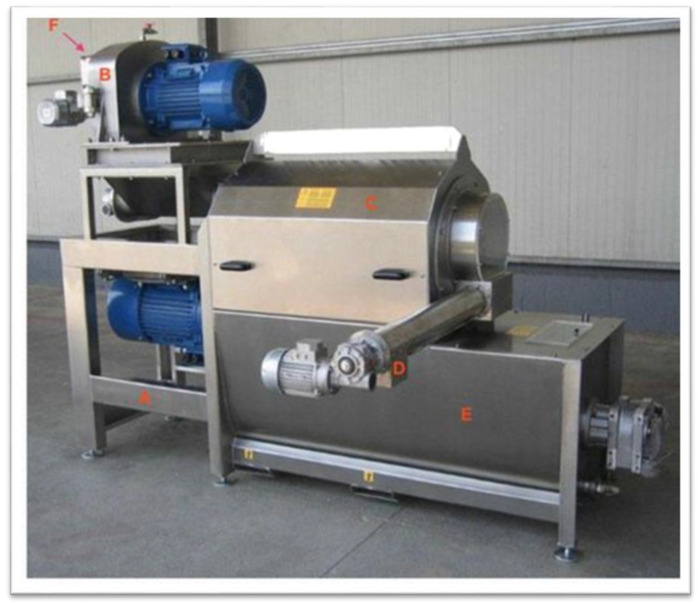
Partial destoner machine proposed by Romaniello, Leone and Tamborrino [16]. Legend: A: chassis; B: mechanical crusher; C: destoner; D: cochlea for pits extraction; E: malaxing; F: olive feeding.

**Table 1 foods-11-01479-t001:** Effects of destoning technology on virgin olive oil phenolic compounds.

Summary and Results	References
Total phenols (mg/Kg) of virgin olive oils obtained from destoned and control (whole fruit) pastes were evaluated at time 0 and after 12 months of storage at room temperature (25 °C). Oils of destoned olive pastes had a content of 355 at the time 0 and 195 mg/Kg after 12 months, vs. oils of whole fruit olive pastes, with values of 345 at the time 0 and 150 mg/Kg after 12 months. Destoning process consented in part to remove peroxidase activity in the pastes, improving the concentration of the hydrophilic phenols in the oils, and their oxidative stability.	Servili et al., 2004 [59]
The authors studied the effect of olive stone removal before processing on the content in secoiridoids and the antioxidant activity of monovarietal extra virgin olive oils. Results showed that destoning increased the total secoiridoids and the antioxidant activity of oils (up to 3.5 times). The study also indicated that these effects depended on variety, assuming that the influence of stone removal was associated with endogenous enzymes.	Lavelli and Bondesan, 2005 [50]
The study compared the phenolic compounds of 16 fresh commercial samples of extra virgin olive oil derived from both stoned and whole fruits. For almost all the samples from stoned fruits, higher concentrations of phenolic compounds were found in agreement with their higher antioxidant capacity. Coratina *cv* showed values of 120 vs. 52.4 mg/L for 3,4-DHEA-EDA in oils obtained from stoned and whole fruits respectively.	Mulinacci et al., 2005 [51]
The research evaluated the quality of virgin olive oils obtained by Coratina *cv* using de-stoner for the olive paste preparation in comparison to the use of a traditional mill. The destoning process caused an increase in the total phenol content of samples.	Amirante et al., 2006 [7]
The influence of destoning technology on minor components and antioxidant activity in two extra virgin olive oils of Bosana *cv*, processed with a two-phase decanter, was investigated. Destoned oils showed great stability and, consequently, had a longer shelf-life than whole fruits oils. During storage, total phenol content was very similar in both oil samples.	Del Caro et al., 2006 [3]
The study reported the effect of fruit destoning on the virgin olive oil phenolic profile determining whether olive seed plays any role in the phenolic content of olive oils. The results showed that increases of about 25% of the total phenolic compounds in oils obtained from de-stoned olive fruits in three Spanish cultivars (Picual, Manzanilla and Hojiblanca) were observed. In fact, olive seeds have been found to contain a high level of peroxidase activity (72.4 U g (−1) FW), responsible for phenols degradation.	Luaces et al., 2007 [47]
The authors observed that removal of the olive stone from the corresponding oils shows a considerable increase in the phenolic fraction, especially the secoiridoid derivatives such as the dialdehydic forms of elenolic acid linked to (3,4 dihydroxyphenyl)ethanol and (phydroxyphenyl)ethanol (3,4-DHPEA-EDA and *p*-HPEA-EDA, respectively) and the isomer of the oleuropein aglycon (3,4-DHPEA-EA) whereas no significant variations of lignans are observed.	Servili et al., 2007 [11]
The study evaluated the oils obtained from destoned olives (Gentile di Chieti, Caroleo, and Coratina cultivars) compared to those with traditional extraction. The destoning has made it possible to obtain highly nutraceutical oils, with a higher content of hydrophilic biophenols. Coratina *cv* showed the highest content of secoiridoids (56 vs. 44 mg/Kg in the destoned and traditional samples, respectively).	Ranalli et al., 2007 [60]
The authors evidenced the effect of the destoning technique on the concentration of bioactive compounds. They found a significant increase in the content of oxidized oleuropein and ligstruside derivates in two oils obtained by Leccino *cv*, destoned vs. stoned. The destoning technique ensures a higher concentration of biophenols, and also richer quantity in α- and γ-tocopherol and in α- and γ-tocotrienol, considering other important bioactive compounds. In particular, oleuropein aglycon, dialdehydic form (9.2 vs. 4.3 mg/Kg in destoned and stoned samples respectively), and ligstroside aglycon, dialdehydic form (24.1 vs. 10.0 mg/Kg in destoned and stoned samples respectively).	Ranalli, A. and Contento, S. 2010 [24]
Antioxidant compounds of extra-virgin oils from Coratina *cv* were evaluated. The total phenolic content of extra-virgin oils was found to be higher in destoned samples (450.7 mg/Kg), versus those from whole olives (338 mg/Kg). Concerning the study of the phenol compounds, destoning led to higher amounts of (+)-1-acetoxypinoresinol and 3,4-DHPEA-EA.	Gambacorta et al., 2010 [10]
This research investigated the effect of stone removal before processing on the antioxidant properties of extra virgin olive oil from Cerasuola *cv*. The amounts were 2.65 and 1.53 μmol GA/g polar extract for fractions from destoned and non-destoned respectively.	Restuccia et al., 2011 [21]
The destoning and malaxation in nitrogen atmosphere on oxidative stability of extra virgin olive oil from olives of Edremit yaglik *cv*. were evaluated. Samples were processed, with or without stones, in nitrogen or air atmosphere. Results have shown that the oils destonated and malaxed in nitrogen flush had a higher total phenols content than those obtained, with the same conditions but not destoned (328 vs. 282 mg/Kg respectively).	Yorulmaz et al., 2011 [29]
The paper reports bionutritional value of destoned (vs. whole) virgin olive oil from Olivastra di Seggiano *cv*. The authors investigated from 2008 to 2010 and showed that removal of the stone from the fruit before processing enhanced the high-quality level of oil, by increasing the biophenols. Concentrations of total oleuropein derivatives were 128.32 vs. 109.11 mg/Kg tyrosol in destoned and whole samples respectively.	Ranalli et al., 2012 [26]
The effects of olive pitting and variety (Greek varieties Koroneiki and Megaritiki) were investigated on the phenolic content of olive oil. The phenols of the pitted olive oils were higher than the whole olive oils in both varieties. The total phenol content of Koroneiki pitted olive oils was 303.45 vs. 226.49 mg/Kg in destoned samples and whole samples respectively.	Katsoyannos et al., 2015 [20]
The authors analyzed phenols and terpenoids in two cultivars Arbequina and Picual after fruits destoning. Destoning has been demonstrated to have different effects for cultivars and especially on secoiridoid derivatives. When olive fruits were destoned concentration of secoiridoids decreased in the Arbequina oil, while it increased in Picual oil	Criado-Navarro et al., 2021 [61]

**Table 2 foods-11-01479-t002:** Effect of destoning technology on virgin olive oil volatile compounds.

Summary and Results	References
The authors observed that the quantitative composition of volatiles deriving from the lipoxygenase pathway was influenced by the olive fruit stones. Volatile compounds of oils obtained from de-stoned olives of Coratina *cv* had a greater accumulation of C6 metabolites than oils extracted by the whole fruits. The sum of all C6 compounds, expressed as ppm, was 54.4 and 33.7 in destoned and whole samples, respectively.	Angerosa et al., 1999 [5]
An investigation on volatile compounds of virgin olive oils Coratina *cv* obtained from the de-stoner olive paste in comparison to the traditional stone mill was conducted. Data showed that de-stoned oils had a higher amount of C5 and C6 volatile compounds, especially, trans-2-hexenal (185.4 vs. 110.8 mg/Kg in destoned and whole samples, respectively) and cis-3-hexen-1-ol (4.8 vs. 8.6 mg/Kg in destoned and whole samples).	Amirante et al., 2006 [7]
The study investigated the effect of stoning removal on the volatile compounds in pulp and seed from Frantoio and Coratina olive cultivars. Data showed that for both the studied cultivars, the amount of the C6 unsaturated aldehydes, such as trans-2-hexenal was higher in the crushed pulp, while the crushed seed was richer in C6 unsaturated alcohols.	Servili et al., 2007 [11]
Destoned olives from Gentile di Chieti, Caroleo and Coratina *cv* were processed in confront with traditional extraction. The de-stoned oils showed higher amounts of pleasant volatiles, such as green aromas C6 unsaturated/saturated aldehydes, C6 alcohols, C6 esters, and C5 compounds. Moreover, in de-stoned samples, two new volatiles (α-copaene and α-murolene) were present.	Ranalli et al., 2007 [60]
The authors analyzed the influence of stone removal on volatile compounds in extra virgin olive oils obtained from Carolea, and Ottobratica cultivars. Data indicated that the oils obtained from destoned olives by the two morphologic different varieties had a greater content of C5 and C6 volatile compounds, compared to that obtained from whole olives, demonstrating that this characteristic was varietal independent.	Runcio et al., 2008 [65]
Composition of volatile fraction in destoned and whole Nocellara del Belice *cv* olives was reported. Results suggest that destoning samples had higher concentrations of C5 and C6 volatile, responsible for the pleasant aromatic green notes in the oil. Unsaturated aldehydes were major metabolites, especially trans-2-hexenal was 425.1 vs. 331.2 mg/Kg in destoned and whole samples, respectively.	Ranalli and Contento, 2010 [24]
The work reported the volatile composition of destoned (vs. whole) virgin olive oil from Olivastra di Seggiano *cv*. Stone removal from the fruit before processing displayed higher levels of C6 green volatiles, such as trans-2-hexenal (963.1 vs. 658.9 mg/Kg in destoned and whole samples, respectively).	Ranalli et al., 2012 [26]

## Data Availability

No new data were created or analyzed in this study. Data sharing is not applicable to this article.

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
