# Peer review of "Extra Virgin Olive Oil from Destoned Fruits to Improve the Quality of the Oil and Environmental Sustainability"

_foods, 2022, doi:10.3390/foods11101479_

Round 1

Reviewer 1 Report

In my opinion, this review is very interesting, the authors gives an overview on the impact of destoning on the olive oil quality. This technology improves the flavour and the amounts of phenolic and volatile compounds.  Also the valorization of destoning products seems a further point in favor of the destoning process; the stoned pomace can be used in animal feeding and the stones can be used as a biomass fuel.

The writing style and English are very good.

Specific comments:

There are many places in the text with double spaces between words, and others without space.

In figure 1 and 2 the authors should explain the traject of the olive destoning. They can use arrows.

The authors should add one or two tables, with references, in which they summarize the impact of destoning on the chemical composition of olive oil (Phenolic compounds, volatile compounds, Sensory characteristics…)

Author Response

We would like to thank the reviewer for providing the valuable comments we received, very much appreciated. We have revised the paper following the suggestions of the reviewer, and we believe that now the quality of the paper is much improved. The authors’ response to the reviewers’ comments is highlighted in yellow as well as with yellow evidence all other corrections were done.

Reviewer 2 Report

  1. Give clearly the aim/scope of the work.
  2. You can give schematic representations of olive oil processes instead of instruments.
  3. You can give more information about olive oil production and its techno-economical properties, and olive oil producer countries. 
  4. Figure 2 - The letters are not seen easily, they should be modified.
  5. What are the POD, K232, K270, LOX on page 3?
  6. You can give more figures and tables, especially for phenolic compounds and volatile compounds.
  7. Clearly check some special characters for example "symbol of degree" in line 234 and "beta" in line 222.
  8. You can discuss the sustainable olive oil production process and its trade over the world. Discuss it with a flow chart. 
  9. How we can improve the quality/quantity of olive oil production? You can clearly discuss them.  

Author Response

(The authors gave the same response as above.)

Reviewer 3 Report

The work is interesting and it seems that the authors almost covered the literature.

I would like to bring to their attention the following articles just in case they may be useful and can be integrated in the manuscript

A. Faizan Tariq, E. Gul, M. Shahid, A. Hamza, H. Jabbar and U. S. Khan, "Design of Olive Pitting Machine," 2021 International Conference on Robotics and Automation in Industry (ICRAI), 2021, pp. 1-6, doi: 10.1109/ICRAI54018.2021.9651337.

Toscano, P.; Cutini, M.; Di Giacinto, L.; Di Serio, M.G.; Bisaglia, C. Development of a Lab-Scale Prototype for Validating an Innovative Pitting Method of Oil Olives. AgriEngineering 20213, 622-632. https://doi.org/10.3390/agriengineering3030040

Rigane, Ghayth, et al. "Change in some quality parameters and oxidative stability of olive oils with regard to ultrasound pretreatment, depitting and water addition." Biotechnology Reports 26 (2020): e00442.

please read carefully the manuscript and revise possible phrases where comparisons are made as e.g.

l.322. Destoned oils had also higher oleocanthal levels, p- HPEA-EDA, compared to whole olive samples (23 vs 21.4 mg/Kg tyrosol).

it is a negligible difference

l. 483 Therefore, a higher polyphenol content in pitted oils would allow adding the health claim on the label, with a positive effect on consumers

it is not certain that this will be achieved if the oil potential is low. However, depitting increases the chance. I would recommmmend appropriate rephrasing

please check for minor spelling corrections

e.g. ref. 24 orgnanic destoned instead of organic destined

Author Response

(The authors gave the same response as above.)

Reviewer 4 Report

The review entitled “A goal of enhancing oil quality and environmental sustainability: extra virgin olive oil from destoned fruits” has been studied and evaluated by this reviewer. In my opinion, the addressed topic is very interesting both for EVOO producers and the scientific community and fits the scope of the journal, but I do not find it suitable for publication in Foods. In the coming paragraphs, I’ll try to give the reasons motivating my decision. I hope my comments encourage authors to invest the time and effort required to prepare a good contribution to be sent for peer review when a minimum quality (in terms of content and form) is reached.

First of all, the English language absolutely needs a revision. There are that many language imprecisions, odd structures, and words wrongly translated that the understanding of the manuscript is almost impossible in some parts. I strongly recommend authors to hire a professional English editing service.

Apart from that, the information included in the manuscript must be reorganized. The structure of the paper seems appropriate, but the content of each section does not always match the title. Moreover, authors cannot limit themselves to give a list of published research works explaining their experimental design and main findings. A review paper must give the reader a deep insight on the addressed topic. It must organise the information available in literature and group papers according whether their hypotheses, methods or findings are in agreement or not. This comment applies to all the manuscript sections.

  1. Introduction

The entire introductory section must be rewritten. Please, find a logical way to order the information, starting with a short paragraph including proper references that describe the agrotechnological parameters affecting VOO composition and quality.

I acknowledge the authors for explaining the main characteristics of both types of pitting machines, but it is indispensable to highlight the different effect of the stone removal before crushing the olive fruits and after that in the obtained oil (throughout the whole contribution). All the benefits related to the absence of the olive seeds and the enzymes they contain (described in the subsequent sections) completely depend on the stoning system. Once the stones are crushed, the seeds are released and co-crushed with the past. The removal of 20-50% of the crushed pits does not inhibit POD or other seed enzymes activity, but the removal of the whole intact stones does. So please, always specify the type of stoning performed in any study reported in this review. The findings cannot be understood without this vital information.

Mentions to pitting machine production companies and brands must be avoided in the paper (it does not add any relevant information for the scientific community). Some ideas reported in lines 64-70 seem personal believes or opinions to me. Assumptions that cannot be proved should not be included in a review like this one.

Although you may give a general depiction of the advantages that extracting EVOO from destoned fruits can bring, please, do not extract conclusions in the Introduction without having referenced the main findings from previous studies.

Lines 78 to 140 must be included in a new section dedicated to production aspects such as extraction efficiency and possible technical difficulties derived from the use of stoned paste. In any case, the reported information must be reorganized (as in the rest of the manuscript). Don’t forget to specify the type of stoning machine used in each research work.

Please, clarify the idea on line 99; what is the relation between the destoning of olive fruits and the link of local traditional olive-growing to its territory?

  1. Fruit characteristics affecting destoning

In line 149, to which kind of stoning machine are authors referring? If the paste is stoned after crushing, how the fruit size can affect the stoning process? Please, clarify and be rigorous on the statements made.

While I understand the relationship between the fertilization, the stone/pulp ratio and the possible effect on the fruit stoning, I do not get the link with the harvesting method. I would remove lines 166-178 and, if keeping line 179, please include a proper reference demonstrating this fact.

  1. Importance of the olives endogenous enzymes.

Summarize the aspects related to pulp enzymes and give illustrative references to check by the interested readers. For example, lines 217-222 should be cut down and the PPO activity description shortened. This section must be focused on seed enzymes and the possible synergistic effects of all the olive fruit enzymes on the obtained oil. Please, give a proper reference for the statement made on lines 233-234.

  1. Effect of destoning technology on phenolic compounds

As already mentioned, the referenced studies in this section need to be grouped and explained in a way that facilitates the understanding of the reader. In fact, to display some data (e.g. stoning machine type and other technological aspects, olive varieties, phenolic content, etc) in tables would be very valuable.

If authors want to include tocopherols in this section, please, change the title to “Effect of destoning technology on minor compounds”. If so, this section could be enriched if they also add information of other families of minor compounds such as pentacyclic triterpenes, sterols or even chlorophylls and carotenoids (if available).

  1. Effect of destoning technology on volatile compounds

As recommended in the previous section, when comparing quantitative amounts of volatile components in oils obtained in different ways, a table summarizing analytical results from comparable studies would be very advisable. That may make it easier for the author to compare data and draw conclusions.

  1. Impact of the destoning on sensory characteristics

Again, studies reporting similar results must be grouped to enhance understandability of the section. Authors should highlight the link between the sensory characteristics and phenolic and volatile compounds described in previous sections.

  1. Future perspectives/8. Conclusions

I would recommend to merge both sections.

Line 482: is there any previous research study reporting that the EFSA criteria was not met by oils obtained in the traditional way (three or two phases) but it was met by oils obtained from the same stoned fruits? The limit is “normally” met by almost every EVOO. Thus, from my point of view, the sentence may be a little bit misleading.

Lines 492-493 are not conclusions, but the idea is interesting and could be placed at the end of the introduction (after line 144) if properly referenced.

Author Response

We would like to thank the reviewer for providing the valuable comments we received, very much appreciated. We agree with most of them. We have revised the paper following the suggestions of the reviewer, and we believe that now the quality of the paper is much improved. The authors’ response to the reviewers’ comments is highlighted in yellow as well as with yellow evidence all other corrections were done. 

Reviewer 5 Report

The manuscript entitled “A goal of enhancing oil quality and environmental sustainability: extra virgin olive oil from destoned fruits”

Authors: Maria Teresa Frangipane, Massimo Cecchini, Riccardo Massantini, Danilo Monarca 

Overview and general recommendation: 

The paper focuses on the influence of destoning on the quality of extra virgin olive oil. The main conclusion is that destoning is one of the most significant advantages for the improvement of the oil qualitative traits and the system's sustainability. This knowledge can provide valuable information about the olive oil current processing and extraction technologies with environmental respect and the excellent characteristics of olive oil. The manuscript's subject is very interesting, and the topic addressed is of great interest for the potential readers of the journal and fits within its scope. The manuscript is prepared professionally. It includes a well-written abstract and an good introduction that justifies the research undertaken. The introduction points to the deficiencies in the literature on the subject. The aim is clearly defined. The discussion of the results is well prepared. The conclusions are well-defined. The illustrative material is poor. In my opinion, the manuscript is suitable for publication in this journal after minor additions.

Below I give my concerns, that need revision. 

  • In my opinion, the title is subject to modification because the phrase "A goal of ..." is not very good here.

Introduction

  • At first, it would be useful to know how popular the technology of pitted olive oil is. What is the percentage of the oil market? In my opinion, it is on the margins of the market. I request information.
  • Next to photo 1, it would be good to show the interior of this device. When viewed from the outside, it is similar to the others used in the process line.
  • “Titouh, Mazari & Meziane [27] reported”, “Guermazi, Gharsallaoui, Perri, Gabsi & Benincasa [28]…” In my opinion, the record of the cited items should be different: Guermazi et al.
  • „As expected, destoning carried out a slight decrease of about 1.1 percent point of the oil yield compared to the whole fruits.” Is such a change significant? is it not within the bounds of error?
  • “Lower oil yield caused by de-stoning” Please provide more information, how much lower is the yield of olive oil? more research, more data. One - 1.1% is unsatisfactory.

Conclusions

  • Conclusions require redrafting, some content should be placed above in the chapter on felted compounds “the Commission Regulation no. 432/2012 a series of claims regarding the advantages of constituents in foods having a biological effect, such as olive oil polyphenols. “The claim may be used on labels, only for olive oil which con- tains at least 5 mg of hydroxytyrosol and its derivatives (e.g. oleuropein complex and tyrosol) per 20 g of olive oil”

Tables, figures

  • There is no number or graphic representation of the collected data, e.g. the content of phenolic compounds, volatile compounds.

References

  • References require standardization, sometimes a DOI number is given, sometimes not, why? Latin names should be italicized.

Author Response

(The authors gave the same response as above.)

Round 2

Reviewer 2 Report

No comments have been made.